# Protective Effects of Flavonoids against Alzheimer’s Disease: Pathological Hypothesis, Potential Targets, and Structure–Activity Relationship

**DOI:** 10.3390/ijms231710020

**Published:** 2022-09-02

**Authors:** Jiao Li, Min Sun, Xiaodong Cui, Chen Li

**Affiliations:** 1School of Life Science, Shanxi University, Taiyuan 030006, China; 2Institute of Biotechnology, Shanxi University, Taiyuan 030006, China

**Keywords:** flavonoids, Alzheimer’s disease, pathological hypothesis, targets, structure-activity relationship

## Abstract

Alzheimer’s disease (AD) is a neurodegenerative disease with high morbidity and mortality, for which there is no available cure. Currently, it is generally believed that AD is a disease caused by multiple factors, such as amyloid-beta accumulation, tau protein hyperphosphorylation, oxidative stress, and inflammation. Multitarget prevention and treatment strategies for AD are recommended. Interestingly, naturally occurring dietary flavonoids, a class of polyphenols, have been reported to have multiple biological activities and anti-AD effects in several AD models owing to their antioxidative, anti-inflammatory, and anti-amyloidogenic properties. In this review, we summarize and discuss the existing multiple pathogenic factors of AD. Moreover, we further elaborate on the biological activities of natural flavonoids and their potential mode of action and targets in managing AD by presenting a wide range of experimental evidence. The gathered data indicate that flavonoids can be regarded as prophylactics to slow the advancement of AD or avert its onset. Different flavonoids have different activities and varying levels of activity. Further, this review summarizes the structure–activity relationship of flavonoids based on the existing literature and can provide guidance on the design and selection of flavonoids as anti-AD drugs.

## 1. Introduction

Alzheimer’s disease (AD), commonly known as senile dementia, is the main type of dementia and is an age-related neurodegenerative disease. The incidence of AD gradually increases with age, and the incidence rate can be as high as 50% in individuals over 85 years old [1]. With the aging of the global population, the number of patients with AD inevitably increases. According to the statistics reported by the World Health Organization, at the end of 2019, more than 50 million people worldwide suffered from AD. The total number of patients with AD globally is estimated to reach 82 million in 2030 and 152 million in 2050. In 2015 alone, the global treatment and care costs associated with AD were USD 957.56 billion, and this figure is estimated to reach USD 2.54 trillion in 2030 and USD 9.12 trillion in 2050 [2]. Despite this significant public health issue, only five medical treatments have been approved for the management of AD. After nearly 20 years, the United States Food and Drug Administration approved aducanumab (Aduhelm), a new drug, for the treatment of AD on 7 June 2021. Among the therapies approved for AD in the United States, this is the first and the only currently used drug to treat AD and not simply alleviate its symptoms. However, current evidence on the effectiveness of Aduhelm is not strong.

At present, there is no clear conclusion on the pathogenesis of AD, but it is generally believed that AD is a disease caused by multiple factors. These factors work synergistically to cause the occurrence and development of AD. It is unlikely that targeting a single change will be effective for disease treatment. Thus, compounds with multiple biological activities that can impact various AD-associated changes that contribute to disease development and progression are needed. An excellent source for multitarget compounds is the original pharmacopeia—plants. Plants can synthesize a huge array of compounds called secondary metabolites, which are not required for plant growth. Among these plant-derived secondary metabolites, flavonoids have been specifically highlighted in several epidemiological studies for their potential role that benefits the prevention of AD. For example, a Mediterranean diet rich in flavonoids might slow age-related cognitive decline and reduce the risk of dementia [3]. Population studies on the correlations between flavonoid intake and dementia in 23 developed countries found that increased consumption of dietary flavonoids, especially flavonols, is associated with lower rates of dementia in the population of those countries [4]. Consistent with these results, a large cohort study based on a random sample of 1367 subjects aged 65 years and higher found that the intake of foods rich in flavonoids is inversely correlated with the risk of the incidence of dementia [5]. Although direct clinical evidence is lacking, the potential effect of flavonoids in the management of AD has been analyzed in some reports. Teles et al. [6] systematically reviewed the preclinical evidence of flavonoids in treating AD. Hole et al. summarized the progress and adverse effects of flavonoids in treating AD [7]. Uddin et al. [8] elaborated on the therapeutic promise of flavonoids in the management of AD through various signaling pathways. Hongjun Xia et al. [9] reviewed the metabolism of flavonoids and their bioactive metabolites and the underlying mechanism to reverse AD pathology. In this review, we summarize and discuss the existing hypotheses in the pathogenesis of AD, including THE neglected “the infectious theory”. Accordingly, we explored the potential mode of action of flavonoids used in the management of AD and the mechanisms underlying their activities, including experimental evidence AND potential targets. Further, the potential benefits of many individual flavonoids such as catechins [10], quercetin [11], apigenin [12], nobiletin [13], fisetin [14], and chrysin [15] in treating AD have also been described in some reviews. These papers show different flavonoids have different activities and levels of activity. Thus, in this review, we summarize the structure–activity relationship of flavonoids based on the existing literature.

## 2. AD and Current Hypotheses on AD

AD is a progressive neurodegenerative disorder. With the development of AD, the memory of affected individuals declines, and their cognitive function is impaired. Patients eventually develop dementia and lose their self-care ability. AD was first reported by Alois Alzheimer in 1907, and the disease was named after him [16]. Alois Alzheimer described two typical pathological injuries in the brains of patients with AD: neurofibrillary tangles (NFTs) in nerve cells and the formation of senile plaques outside the nerve cells. The main component of NTFs is the abnormally hyperphosphorylated microtubule-associated protein (tau), which exists in the form of aggregated double-helical filaments. Senile plaques are mainly formed by abnormal folding and aggregation of amyloid-beta (Aβ). Although these two pathological occurrences have always been used as diagnostic indicators of AD, many controversies about the pathogenesis of AD continue to exist [17]. Several overlapping mechanisms have been proposed to explain the underlying pathology of AD, and both current and future potential treatments are based on the modification of these mechanisms.

## 3. Amyloid Cascade Hypothesis

The “amyloid cascade hypothesis” is the most classic hypothesis elucidating the pathogenesis of AD. This hypothesis suggests that the deposition of Aβ protein, the main component of plaques, is the causative agent of AD [18]. Aβ is a polypeptide consisting of about 42 amino acids, and it is formed by the sequential cleavage of Aβ precursor protein (APP) by β-secretase (β-amyloid cleaving enzyme 1, BACE1) and γ-secretase, which is also found to be the main cleavage pathway of APP in individuals with AD [19,20]. Aβ has a strong self-aggregation propensity and can thus easily form the Aβ oligomer and deposits, all of which are cytotoxic [21,22]. They can trigger a cascade leading to synaptic damage and neuron loss and ultimately to the pathological hallmarks of AD, which are amyloid plaques and NFTs consisting of hyperphosphorylated tau protein [23].

The “amyloid cascade hypothesis” cannot comprehensively explain the neuronal damage in AD due to several reasons [24]. For example, amyloid deposition can be found in cognitively normal elderly subjects, and treatments that reduce Aβ load do not modify the progression of AD both in animal and human models [25]. Nevertheless, many AD research communities still believe that Aβ is the most basic component for the occurrence of AD and acts as a trigger for other pathogenic factors [26,27] Thus far, aducanumab is the first Aβ-targeting drug that has received approval for the treatment of AD.

## 4. Tau Hypothesis

The tau hypothesis supports that the principal causative substance of AD is tau protein. Tau is a phosphoglycerate protein that contains 2–3 phosphate groups and is expressed in neurons that normally function in the stabilization of microtubules in the cell cytoskeleton [28]. The tau protein content in the brains of patients with AD is about 3–4 times that in healthy individuals. Moreover, tau protein is abnormally hyperphosphorylated, and each molecule of tau protein in patients with AD can contain 5–9 phosphate groups [29]. Hyperphosphorylation causes the tau protein to accumulate as NFT masses within nerve cell bodies. This is a major pathological feature of AD in addition to Aβ deposition. These tangles then aberrantly interact with cellular proteins, preventing them from executing their normal functions. Clinically, the degree of phosphorylation of tau protein has a positive correlation with the severity of AD [30]. The tau hypothesis also suggests that tau lesions occur prior to Aβ accumulation.

## 5. Oxidative Stress Hypothesis

Oxidative stress is another common pathological feature of AD [31]. Compared with other aerobic tissues and organs, neurons of the brain are more prone to oxidative stress due to several reasons. First, the brain, a component of the central nervous system, consumes more oxygen, which leads to increased ROS production [32]. Second, the intrinsic antioxidant defense of the neurons of the brain is weak [33] because they have low levels of antioxidant enzymes or compounds, such as superoxide dismutase, peroxidase, and glutathione. Additionally, antioxidants in other body parts cannot easily cross the blood–brain barrier (BBB). Moreover, the cell membranes of neurons in the brain contain high levels of unsaturated fatty acids, which are prone to react with free radicals through peroxidation reactions [34,35]. ROS accumulates in the neurons of the brain, causing neurotoxicity and neuronal synaptic dysfunction. In AD, Aβ aggregation, tau phosphorylation, metal ion accumulation, and inflammatory response can stimulate excessive production of ROS and damage the antioxidant defense ability, causing oxidative stress [36,37,38].

## 6. Inflammation Hypothesis

Over the past decade, evidence has indicated that reactive gliosis and neuroinflammation are the hallmarks of AD [39,40,41,42]. Immunohistochemical experiments have shown that inflammatory markers such as acute phase proteins, complement, and activated microglia are detected in the brain tissues of patients with AD. Proinflammatory and anti-inflammatory cytokines (eotaxin, interleukin (IL)-1ra, IL-4, IL-7, IL-8, IL-9, IL-10, IL-15, granulocyte colony-stimulating factor, monocyte chemotactic protein 1, platelet-derived growth factor, tumor necrosis factor (TNF)-α) have also been found to increase in the serum and brain tissues of patients with AD [43,44,45,46,47]. Upregulation of anti-inflammatory cytokine levels plays a “protective” role in AD [46]. As an inflammatory-stimulating factor, Aβ protein in the brains of AD patients can activate astrocytes and microglia to release inflammatory factors with strong neurotoxicity, including nitric oxide, TNF, and complement Cl and C3 [48]. Overexpression and the complex interaction of these factors can lead to inflammation and promote the generation of free radicals, which can damage neuronal cells [49]. Apart from their direct neurotoxic effects, proinflammatory cytokines can activate β-secretase to promote Aβ deposition [50,51]. Several studies in mouse models have shown that the level of amyloid deposition is increased under inflammatory conditions [52].

## 7. Cholinergic Hypothesis

The cholinergic hypothesis has been proposed to describe the pathogenesis of AD earlier and is one of the currently and widely accepted hypotheses [53]. Acetylcholine (ACh) is an important neurotransmitter used by cholinergic neurons for many critical physiological processes such as cognition and memory [54,55]. Cholinergic neurons in the normal basal forebrain can synthesize a large amount of ACh. However, in the brains of patients with AD, a large number of cholinergic neurons are lost, and the activity of choline acetyltransferase is reduced, resulting in a significant reduction in the synthesis, storage, and release of ACh. Cholinergic neuronal damage is considered a critical pathological change that is correlated with cognitive impairment observed in AD. Aβ accumulation, hyperphosphorylation of tau protein, oxidative stress, inflammation, and metal imbalance cause cholinergic neuronal damage. The first drug approved for the treatment of AD was tacrine, a cholinesterase inhibitor. However, the drug was withdrawn from the market in 2012, as it resulted in severe side effects. Thus far, acetylcholinesterase (AChE) inhibitors are the most widely used drugs in the clinical treatment of AD [56,57,58,59,60].

## 8. Metal Ion Hypothesis

Studies have shown that the balance of metal ions, especially zinc, iron, and copper, in patients with AD is disrupted, causing them to be redistributed in the brain. Post-mortem analyses of patients with AD have revealed the accumulation of copper, iron, and zinc to be 5.7, 2.8, and 3.1 times higher, respectively, than that observed in normal brains [61,62]. The breakdown of ion distribution balance can lead to an imbalance of metal protein. The metabolic disorder of metal ions in the cerebral cortex and hippocampus can cause the accumulation and aggregation of Aβ. Free metal ions with reducing properties (such as Fe^2+^ and Cu^+^) can also be combined with reactive oxygen ions, which can then cause neuronal damage and eventually lead to neurodegeneration [63,64].

## 9. New Pathway of AD—The Infectious Theory

The idea that infectious agents in the brain play a role in the pathogenesis of AD was proposed almost 30 years ago. However, this theory was largely disregarded by the AD research community for many years owing to the lack of substantial evidence. Several recent discoveries have reignited interest in the infectious theory of AD. Researchers represented by Robinson and Bishop have suggested that AD is the result of viral and bacterial infections, such as herpes simplex virus type 1 (HSV-1) [65], *Porphyromonas gingivalis* [66], and *Borrelia burgdorferi* [67]. Aβ is an antimicrobial peptide that may aggregate in response to the presence of infectious agents in the brain [68,69,70,71,72]. This hypothesis is supported by various findings such as the identification of microbial DNA in Aβ plaques [73], the stimulation of Aβ production by infectious pathogens [71,74], and the discovery of antimicrobial activity of Aβ [75].

The above hypotheses indicate different aspects of the pathogenesis of AD; however, the exact mechanism is still unclear. Currently, it is generally believed that AD is a disease caused by multiple factors that can work synergistically and form a feedback loop mechanism (Figure 1).

## 10. Flavonoids

At present, the clinical drug therapy for AD is mainly divided into two categories. One is AChE inhibitors, represented by donepezil, and is suitable for symptomatic relief of mild-to-moderate AD. Another kind is N-methyl-D-aspartate receptor (NMDA) antagonists, represented by memantine, and is suitable for symptomatic relief of severe AD. AChE inhibitors can inhibit the activity of AChE, which leads to an increase in neural ACh levels and improves cognition and memory. NMDA antagonists can inhibit the overactivation of NMDA receptor. The overactivation of NMDA receptors is an important factor leading to cognitive impairment in AD. However, neither class of drugs slowed the progression of AD. They can only improve the quality of life of AD patients. At the same time, these drugs have some side effects, such as lead to vomiting, insomnia and diarrhea, which reduce the compliance of patients with such drugs. Therefore, it is urgent to develop safe and effective pharmacological treatments for AD treatment

Flavonoids are a class of secondary metabolites that have attracted attention from the pharmaceutical industry because of their versatile therapeutic properties. In addition to their medicinal values, flavonoids constitute a part of the human diet due to their abundance in vegetables, fruits, seeds, and beverages (such as coffee, tea, and red wine). More than 9000 different flavonoids have been identified, all of which are divided into six subclasses based on their molecular structure (Figure 2). The basic configuration of flavonoids is C6-C3-C6. Based on the structure of the C ring attached to the A and B rings and the unsaturation of the C ring, flavonoids can be categorized into flavones, flavanones, isoflavones, flavonols or flavan, and anthocyanins, whereas the nonflavonoid group comprises phenolic and lignan acids [76,77]. The structures of flavonoids discussed in this review are summarized in Table 1.

In plants, flavonoids have long been thought to be synthesized in specific parts and are responsible for the color and aroma of flowers. They are used to attract pollinators to aid in seed dispersion, spore germination, and seedling growth and development [78]. Flavonoids play an important role in plants in resisting biological and abiotic stress while acting as ultraviolet filters [79,80]. Moreover, flavonoids have been shown to have a positive impact on human and animal health, and the current interest in these compounds is related to disease therapy and chemoprevention [81]. In both in silico analysis (Table 2) and various models (Table 3), flavonoids have exhibited good biological activities through different pathways, and owing to their antioxidant, anti-inflammatory, anti-amyloid fibrotic, anti-bacterial, enzyme-regulating, and other capabilities, flavonoids have several biochemical and pharmacological effects in the management of AD (Table 3) and other diseases. Corresponding to the pathogenesis hypothesis of AD summarized above, the potential targets for flavonoids to play a role in AD as well as the structure–activity relationship and relevant experimental evidence for each target are summarized below.

## 11. Flavonoids Exert Anti-AD Effects by Affecting Aβ

Although there are many hypotheses about the pathogenesis of AD, Aβ has always been regarded as the most important substance leading to AD. Currently, the two passive immunizations against Aβ, bapineuzumab and solanezumab, have failed in their phase III clinical trials; nevertheless, there are still many anti-AD studies based on the “amyloid cascade hypothesis”.

Vepsäläinen et al. found that anthocyanin-rich bilberry or blackcurrant extracts can decrease the levels of C-terminal fragment of APP in the cerebral cortex of APP/PS1 mice (APdE9) and reduce behavioral abnormalities associated with AD [140]. It has also been reported that the oral administration of grape-derived polyphenols, composed of catechin and epicatechin in monomeric (~8%), oligomeric (75%), and polymeric forms (~17%), for five months can prevent Aβ oligomerization and attenuate AD-type cognitive impairment in a transgenic mouse model of AD of Tg2576 and APP/PS1 mice [141,142]. A study on the long-term administration of *Ginkgo biloba* extracts, including flavonols (quercetin, kaempferol, isorhamnetin) and terpenelactone, has also shown that the extracts can considerably decrease the levels of APP in the cortex of Tg2576 mice [143]. Among flavonoids, epigallocatechin gallate (EGCG) from tea is the most widely and thoroughly studied compound. EGCG has been found to reduce Aβ levels in the brain by controlling APP processing [144,145]. Moreover, EGCG can reduce the formation of β-sheet-rich amyloid fibrils by directly binding to the native, unfolded polypeptides [146]. EGCG can also degrade large Aβ fibrils into smaller ones, which are amorphous protein aggregates that are nontoxic [147]. Other classes of flavonoids show anti-amyloidogenic features such as EGCG. In vitro studies show that morin, quercetin, kaempferol, fisetin, chrysin, (+)-catechin, (−)-epicatechin, and myricetin in particular can exert anti-amyloidogenic activity reversibly and specifically by binding to the amyloid fibril structure of Aβ instead of to the monomers of Aβ [148,149,150]. As shown in Table 3, some flavonoids also showed anti-amyloidogenic features by different pathways in AD models. For example, wogonin can attenuate the amyloidogenic pathway by decreasing the levels of β-secretase, APP β-C-terminal fragment, phosphorylated tau, and the extent of Aβ aggregation in a 3xTg-AD mouse model [124].

In summary, flavonoids exert their antiamyloidogenic effects by (1) decreasing the levels of APP; (2) reducing Aβ production by controlling APP processing; (3) protecting against the aggregation of Aβ; and (4) degrading large Aβ fibrils into smaller ones (Figure 3). Three structural characteristics of natural flavonoids have been proposed to explain their inhibitory activity against the aggregation of Aβ in vitro. These structural characteristics include (1) catechol-type flavonoids, which can form Michael adducts with the side chains of Lys16 and Lys28 in monomeric Aβ_42_ through flavonoid autoxidation, and (2) non-catechol-type flavonoids with planarity due to α,β-unsaturated carbonyl groups that can interact with the intermolecular β-sheet region in Aβ_42_ aggregates, especially aromatic rings such as those in Phe19 and Phe20 [151].

## 12. Flavonoids Exhibit Anti-AD Effects through Anti-Inflammatory Activity

Inflammation is one of the main causes of AD despite the fact that neuroinflammation is usually thought to be the result of AD pathogenesis [152]. Flavonoids have been shown to exhibit anti-inflammatory activity in various in vitro and in vivo models [153]. Furthermore, some reports about flavonoids alleviating AD symptoms through anti-inflammatory effects are presented in Table 3. The effects of flavonoid-rich foods on inflammation in humans have also been reported. For example, supplementation with flavanol-rich black tea and cocoa tablets for 4 weeks can increase P-selectin plasma levels [154,155], and the consumption of black tea for 6 weeks may reduce CRP levels [156]. Estruch et al. reported the reduction of the plasma levels of fibrinogen, IL-1α, CRP, vascular cell adhesion molecule-1, and intercellular adhesion molecule-1 after 4 weeks of red wine consumption [157]. Carmela Spagnuolo et al. reviewed the role of inflammation in neurodegenerative diseases, highlighting the potential therapeutic effects of flavonoids as anti-inflammation agents [158].

The anti-inflammatory effect of flavonoids can be attributed to the inhibition of the synthesis and activity of different pro-inflammatory mediators such as eicosanoids, cytokines, adhesion molecules, and C-reactive protein.

Firstly, flavonoids can inhibit the synthesis and activity of different pro-inflammatory enzymes, which can include cyclooxygenase-2 (COX-2), lipoxygenase (LOX), and inducible NO synthase (iNOS). COX-2 is an inducible enzyme expressed only after being stimulated by an inflammatory stimulus; it plays a role in prostaglandin synthesis for the induction of inflammation [159]. LOX catalyzes the oxidation of unsaturated fatty acids such as arachidonic acid in white blood cells, which are involved in the inflammatory response [160]. The studies were carried out using in silico methods to explore the binding modes between COX-2 and the commercially available flavonoids, including silibinin, galangin, scopoletin, hesperitin, genistein, daidzein, esculetin, taxifolin, naringenin, and celecoxib, and the findings show that these flavonoids have high binding energy [161]. The structure–activity relationship between flavonoids and COX-2 can be summarized as follows. Flavanones and chalcones do not have an inhibitory effect on the COX-2 protein. In flavones, the olefin at C2 and C3 forms a planar structure that is important for COX-2 inhibition. The presence of 5,7-; 6,7-; and 7,8-dihydroxy groups abolish activity, but COX-2 inhibition is observed if a hydroxy group is present in the B ring, such as in luteolin, where the 3′-OH coordinates with Ser353 and the 4′-OH coordinates with Leu352. Among flavonols, the olefin at C2 and C3 forms a planar structure, which is also an important aspect. The presence of the 5,7-dihydroxy group enhances the COX-2 inhibitory effect, and the substituent moiety at C3′ and C4′ also increases COX-2 inhibition as does the carbonyl group at C4. A B ring capable of rotation is also important for activity; thus, morin is an ineffective inhibitor of the COX-2. For flavone glycosides, glycosyl substitution on C3 has minimal effect on the binding of inhibitors and COX-2. Similarly, in silico docking studies have reported the inhibitory effect of various flavonoids against LOX, and the LOX inhibitory activity of the selected compounds was found to be decreased in the order of morin, azelastine, daidzein, naringenin, esculetin, galangin, scopoletin, taxifolin, silbinin, and genistein [162]. Moreover the structure–activity relationship among a series of flavonoids concerning 5-LOX inhibition was assessed through a systematic study of the inhibition of the formation of LTB4 in human neutrophils, and the results showed that (1) the presence of a hydroxyl group in the flavonoid molecule is not essential; (2) a catechol arrangement reinforces the inhibitory effect; (3) in the presence of a catechol arrangement, the inhibitory potency inversely correlates with the number of hydroxyl groups; and (4) a 2,3-double bond in the C ring strengthens the inhibitory effect [163]. Nitric oxide (NO) produced by iNOS is one of the inflammatory mediators. In lipopolysaccharide (LPS)-treated RAW cells, flavonoids such as apigenin, genistein, and kaempferol have been shown to inhibit NO production by reducing iNOS expression and do not directly inhibit iNOS enzyme activity [164]. The structure–activity relationship of flavonoids with respect to iNOS expression inhibition is as follows. A C-2,3 double bond and 5,7-dihydroxyl groups in the A ring might be important for the inhibitory activity of flavonoids. In addition, the 8-methoxy group in the A ring and the 4′- or 3′,4′-vicinal substitutions in the B ring may favorably affect inhibitory activity, whereas 2′,4′-(meta)-hydroxy substitution in the B ring (morin) abolish inhibitory activity. The 3-hydroxy moiety in the C ring might reduce activity; thus, the inhibition of NO production by flavone derivatives such as chrysin, apigenin, and luteolin is stronger than that achieved using flavonol derivatives such as galangin and quercetin.

Secondly, flavonoids can decrease the expression of different pro-inflammatory cytokines/chemokines including TNF-α, IL-1β, IL-6, IL-8, and monocyte-chemoattractant protein-1 in different cell types such as RAW macrophages, Jurkat T cells, and peripheral blood mononuclear cells. Quercetin and catechins exert a synergistic inhibitory effect on TNF-α and IL-1β to enhance the release of the anti-inflammatory cytokine IL-10 [165]. Genistein has been reported to inhibit IL-1β, IL-6, and TNF-α production in LPS-induced human blood monocytes and RAW cells [166,167]. Silybin and quercetin have been shown to inhibit TNF-α production in LPS-treated RAW cells, and the action of quercetin has been attributed to inhibiting MAPK and AP-1 DNA binding [167,168].

## 13. Flavonoids Exert Anti-AD Effects through Antioxidant Activity

Oxidative stress is one of the direct causes of AD. Almost every class of flavonoids can function as antioxidants. It has been reported that flavones and catechins may be the most powerful flavonoids that can protect from ROS. Oliveira et al. reviewed the antioxidant effect of flavonoids present in *Euterpe oleracea martius* and neurodegenerative disease [77], and the same is true of other plant-derived flavonoids.

Flavonoids can prevent injury caused by free radicals in various ways. One way is the direct scavenging of free radicals. Flavonoids can be oxidized by free radicals, causing the radicals to become more stable and less reactive [169], and this occurs due to the high reactivity of the hydroxy groups in flavonoids. Bors et al. [170] summarized the relationship between the structure of flavonoids and their free radical-scavenging activity and reported that the key determinants for radical-scavenging effect are (1) the presence of a catechol group in the B ring, which has better electron-donating properties and is the target of radicals, and (2) a 2,3-double bond conjugated with the 4-oxo group, which is responsible for electron delocalization.

Another antioxidant mechanism of flavonoids is the suppression of ROS formation by inhibiting enzymes or chelating trace elements that are involved in free-radical production, such as xanthine oxidase (XO), free iron, and copper. Several flavonoids such as quercetin, rutin, baicalein, baicalin, and silibin have been found to inhibit XO activity, which leads to decreased oxidative injury [171,172]. Studies on the structure–function relationship by Cos et al. [172] demonstrated that the flavonoid luteolin is the most potent inhibitor of XO. Free metal ions with reducing properties (such as Fe^2+^ and Cu^+^) are potential enhancers of reactive oxygen species formation, as exemplified by the reduction of hydrogen peroxide with generation of the highly aggressive hydroxyl radical. On the one hand, flavonoids can chelate these metal ions to remove a causal factor for the development of free radicals. For example, quercetin can prevent oxidative injury in the erythrocyte membrane induced by a number of oxidizing agents that cause the release of iron in its free or redox-active forms [173,174,175]. On the other hand, flavonoids form complexes with metal ions to enhance their antioxidant activity [176]. Morin, kaempferol, and quercetin have been shown to form complexes with Cd(II) and exhibit strong antioxidant activity in vitro [177,178]. Rodríguez-Arce et al. focused on the antioxidant activities of flavonoids and the increase in their antioxidant activity upon coordination with metal ions to generate metal-flavonoid complexes [176]. Furthermore, the metal complexes design may offer an efficient approach for the development of potential new drugs for the treatment of AD. However, flavonoids have also been reported to show prooxidant effects by reducing iron and copper ions. These reduced metals can catalyze the production of hydroxyl radicals through Fenton reaction and lipid radicals through the decomposition of preformed lipid hydroperoxides.

Another antioxidant pathway of flavonoids in vitro is that they can upregulate or enhance antioxidant defense. Several common flavonoids such as quercetin, naringenin, myricetin, and apigenin have been found to enhance antioxidant defense by increasing SOD, CAT, and GPx activities in mice and cell models [179,180,181]. In addition, some flavonoids can also activate the synthesis of low-molecular-weight antioxidants, such as glutathione, or regenerate vitamin E from its radical form.

## 14. Flavonoids Exert Anti-AD Effects by Acting as Metal-Ion-Chelating Agents

An imbalance of metal ions in the brain can lead to toxicity that is closely related to AD. Several flavonoids have been shown to have excellent ability to chelate metal ions, as described in the section “Flavonoids exert anti-AD effects through antioxidant activity”. In AD, the chelation of metal ions by flavonoids can reduce not only the generation of oxidative stress but also the metal-ion-induced aggregation of Aβ. Initially, myricetin was found to modulate metal-mediated Aβ aggregation and neurotoxicity in vitro and in human neuroblastoma cells owing to its ability to chelate metal and interact with Aβ [182]. Subsequently, (−)-epigallocatechin-3-gallate was found to influence both metal-free and metal-induced Aβ aggregation [183]. DeToma et al. [184] investigated the interaction and reactivity of isoflavone derivatives (aminoisoflavones) with metal-free and metal-associated Aβ and found that the catechol moiety may also be important for the interaction between ligands and metal-associated Aβ.

Mira et al. [185] investigated the interactions and mechanism of flavonoids with iron and copper ions to study their antioxidant activity. In their study, only flavones (apigenin, luteolin, kaempferol, quercetin, myricetin, and rutin) and the flavanol catechin were found to interact with metal ions. The proposed trace metal-binding sites of flavonoids are the catechol moiety in the B ring; the 3-hydroxyl, 4-oxo groups in the heterocyclic ring; and the 4-oxo, 5-hydroxy groups between the heterocyclic ring and the A ring. Among all sites, the catechol moiety is the main site that contributes to metal chelation [183,184]. This is the reason why quercetin can better chelate metal ions than kaempferol. In the study of Mira et al. [186], all flavonoids that were studied were reported to better reduce copper ions than iron ions, and the flavonoids with better Fe^3+^ reducing activity are those with a 2,3-double bond and possessing both the catechol group in the B ring and the 3-hydroxyl group. The copper-reducing activity seems to depend largely on the number of hydroxyl groups.

## 15. Flavonoids Exert Anti-AD Effects by Inhibiting AChE

AChE is a key enzyme in the central nervous system. Its inhibition can lead to an increase in neural ACh levels, which is one of the therapeutic targets for symptomatic relief of mild-to-moderate AD [187]. Hence, the inhibition of AChE is one of the central focuses of drug development. Three of the five drugs used in the clinical management of AD in recent years are AChE inhibitors. Several flavonoids have been reported to exert AChE inhibitory effects. Noui et al. [188] explored the bioactive compounds of anticholinesterase activities in Ephedra and found that phenolic including flavonoids showed a potent inhibitory effect against AChE. Khare et al. [189] screened and identified of secondary metabolites to exert AChE inhibitory effects in the bark of *Bauhinia variegata* by using molecular docking and molecular dynamics simulation and found dihydroquercetin may act as a good inhibitor for treating Alzheimer’s disease. Khan et al. [190] studied the in vitro inhibitory activity of various flavonoids including quercetin, rutin, kaempferol 3-O-β-D-galactoside, and macluraxanthone against AChE. Their results show that quercetin and macluraxanthone possess inhibitory activity against AChE in a concentration-dependent manner. Additionally, macluraxanthone has higher inhibitory activity than quercetin. Myricetin and dihydroxymyricetin have also been shown to effectively inhibit AChE [191,192]. The structural characteristics of dihydroxymyricetin lead to stronger anti-AChE activity than myricetin [191]. Additionally, it has been reported dihydromyricetin improved cognitive impairments in d-galactose-induced AD mice through the inhibition of AChE [193]. In addition to this, Das et al. [194] found that 5,7-dihydroxy-4′-methoxy-8-prenylflavanone isolated from the leaves of *Artocarpus anisophyllus* Miq. could efficiently inhibit AChE.

## 16. Flavonoids Exert Anti-AD Effects by Inhibiting Bacteria and Viruses

The infectious theory of AD suggests that bacterial and viral infections can induce AD. In fact, some microorganisms, such as *Porphyromonas gingivalis* and HSV, have been confirmed to be linked to AD initiation and progression [65,66]. Herbs that are rich in flavonoids have also been used as antimicrobial agents to treat human diseases for centuries. At present, flavonoids have been recognized for antifungal, antiviral, and antibacterial activities. Quercetin, apigenin, and myricetin are three representative flavonoids, all of which have been reported to possess antibacterial activity by acting on multiple targets [195,196]. Quercetin and myricetin have also been shown to have inhibitory activity against *P. gingivalis*, which is associated with AD [195,196].

Among the antiviral properties of flavonoids, the most studied effect is their anti-HIV property. In vitro studies have shown that several flavonoids can inhibit HIV-1 infection and replication by inhibiting HIV-1 enzymes (HIV-1 reverse transcriptase, HIV-1 integrase, and HIV-1 proteinase). These flavonoids include baicalin, baicalein, myricetin, quercetin, robustaflavone, hinokiflavone, robinetin, and quercetagetin [197,198,199,200]. Flavonoids also have an inhibitory effect on many RNA and DNA viruses such as HSV, respiratory syncytial virus, poliovirus, and Sindbis virus by inhibiting their replication and preventing infectivity [201,202]. Lyu et al. [203] found that flavanols and flavonols potentially have higher anti-HSV activity than flavones. In their study, epicatechin, epicatechin-3-gallate, galangin, and kaempferol showed strong anti-HSV activity, whereas catechin, EGC, EGCG, naringenin, chrysin, baicalin, fisetin, myricetin, quercetin, and genistein showed moderate activity. In those experiments, flavanols and flavonols appeared to be more active than flavones. The anti-*P. gingivalis* and anti-HSV effects of flavonoids should help reduce the risk of AD.

Taken together, the findings indicate that flavonoids can be potential candidates for the multitarget prevention and treatment of AD. They may help slow the progression of AD or delay its onset. These reports suggest that different flavonoids may exhibit anti-AD activities by targeting different targets. for the same target, flavonoids of different structures have different activity, and a catechol group and in the B ring and a 2,3-double bond in the C ring may be important for various activities of flavonoids. Thus, simultaneously using multiple flavonoids or dietary flavonoid supplements may be an effective multitarget approach for the prevention and treatment of AD. However, because some reports present different results for the same flavonoid, it is clear that more experiments and/or clinical trials are needed to confirm the anti-AD effect of flavonoids.

## 17. Conclusions

The pathogenesis of AD involves several mechanisms and factors, including the “amyloid cascade hypothesis”, “tau hypothesis”, “oxidative stress hypothesis”, and “inflammation hypothesis”. Growing knowledge of AD has indicated that these factors work synergistically, forming a feedback loop mechanism that leads to the development of AD. In addition, pathophysiological brain alterations usually occur decades before the clinical signs and symptoms manifest. Consequently, the current treatment strategies for AD should be multitargeted and used for early intervention to jointly prevent disease progression.

Flavonoids are ideal compounds and are in line with the treatment strategies for AD for multiple reasons. First, flavonoids have a wide range of activities that could make them particularly effective for blocking AD-associated pathogenic factors. Second, flavonoids, especially flavonoid glycosides, can cross the blood–brain barrier and directly act on the brain. Third, flavonoids can be isolated from various plants and effectively used to reduce the incidence of age-related diseases, as indicated by studies using diverse animal- and cell-based models. Collectively, flavonoids can be regarded as prophylactics to slow the advancement or to avert the onset of AD. Consuming foods rich in flavonoids, such as by following a Mediterranean diet, is recommended, especially for the elderly. Flavonoids can also be promising candidates as anti-AD drugs.

However, the therapeutic development of flavonoids has been hindered by an ongoing lack of clear mechanistic data that fully consider metabolism and the bioavailability of flavonoids in vivo. There is also a lack of translational research and clinical evidence of these promising compounds. Therefore, further studies are required to address the specific processes by which flavonoids exhibit their neuroprotective effects and to determine the role of flavonoids in humans.

## Figures and Tables

**Figure 1 ijms-23-10020-f001:**
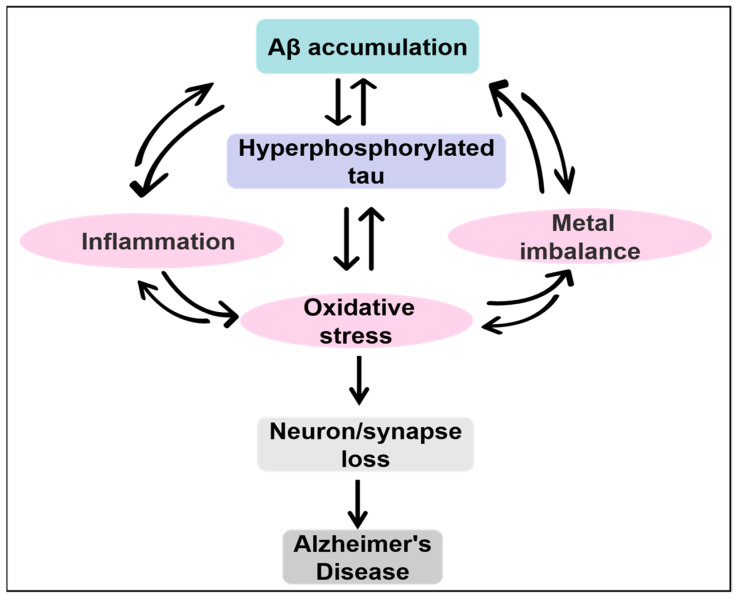
Mechanism of AD pathogenesis. AD is a disease caused by multiple factors, such as Aβ accumulation, hyperphosphorylated tau, oxidative stress, inflammation, metal imbalance, etc. These factors promote one another and form a feedback loop mechanism. Among all the factors, Aβ accumulation and hyperphosphorylated tau are the two main diagnostic indicators of AD and are the main substances that cause the occurrence of AD.

**Figure 2 ijms-23-10020-f002:**
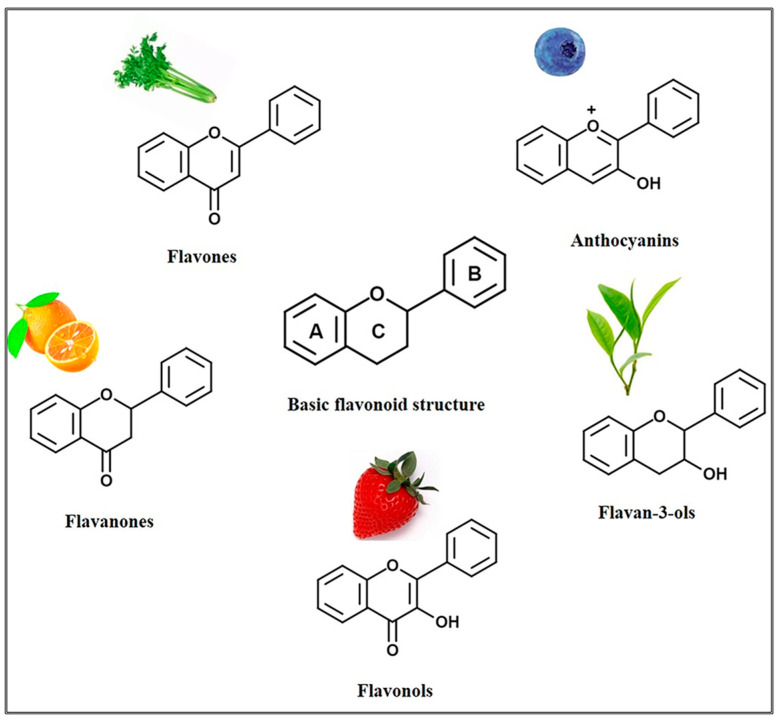
Chemical structures of various classes of flavonoids. Flavonoids have the basic configuration of C6-C3-C6. According to the structure of the C ring attached to the A and B rings and the unsaturation of the C ring, they can be divided into: flavones, flavanones, isoflavones, flavonols or flavan, and anthocyanins.

**Figure 3 ijms-23-10020-f003:**
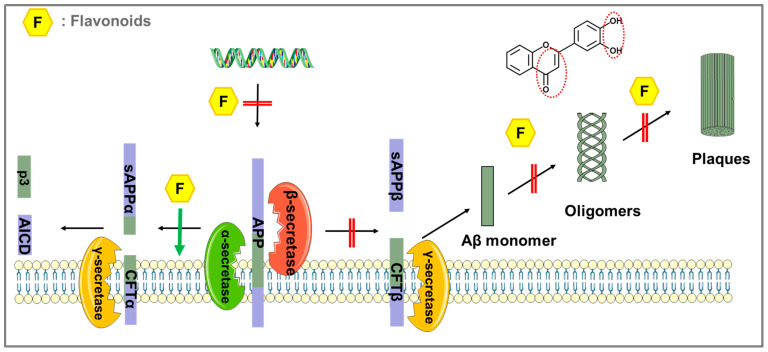
Flavonoids exert anti-AD effects by interacting with Aβ. Flavonoids exert anti-amyloidogenic effects through: (1) decreasing the levels of APP; (2) reducing Aβ production by regulating APP processing; and (3) preventing the aggregation of Aβ. The structural characteristics for inhibitory activity against the aggregation of Aβ include a catechol group and in the B ring and a 2,3-double bond in the C ring. The double red line represents the inhibitory effect, and the green arrow represents the promoting effect.

**Table 1 ijms-23-10020-t001:** Structure of flavonoids in the text.

Catechins 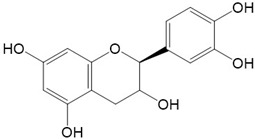	2. Quercetin 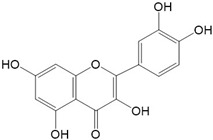
3. Apigenin 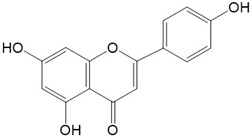	4. Nobiletin 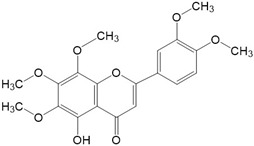
5. Fisetin 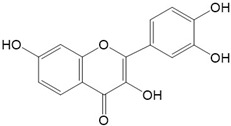	6. Chrysin 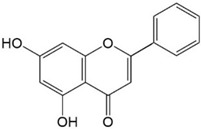
7. (−)-Epicatechin 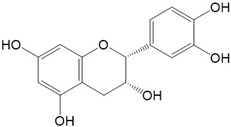	8. Kaempferol 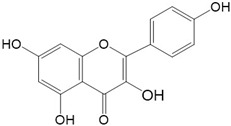
9. Isorhamnetin 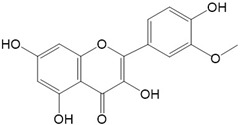	10. Morin 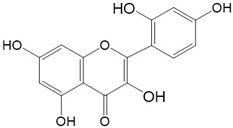
11. Myricetin 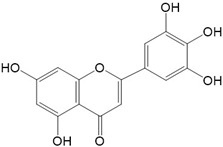	12. Wogonin 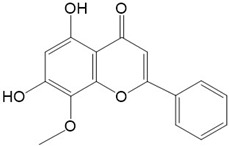
13. Silybin 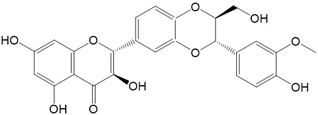	14. Galangin 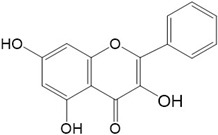
15. Hesperetin 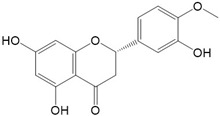	16. Genistein 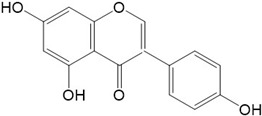
17. Daidzein 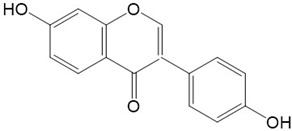	18. Taxifolin 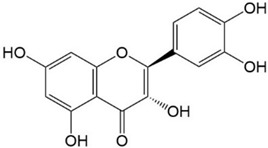
19. Naringenin 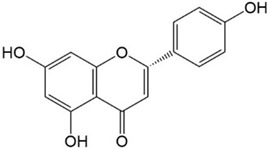	20. Daidzein 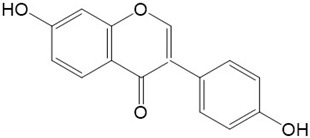
21. Baicalein 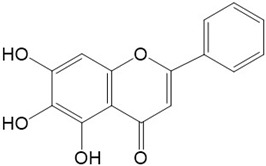	22. Epigallocatechin-3-gallate 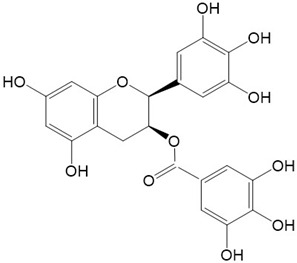
23. Luteolin 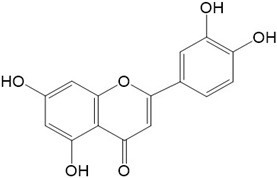	24. Dihydroxymyricetin 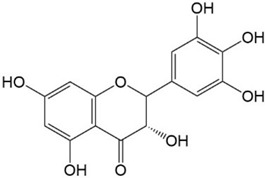
25. 5,7-dihydroxy-4′-methoxy-8-prenylflavanone 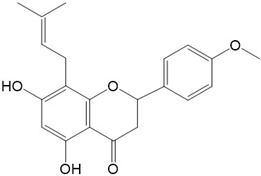	26. Quercetagetin 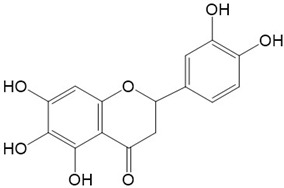
27. Hinokiflavone 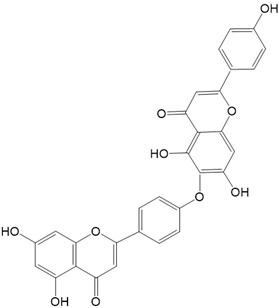	28. Robinetin 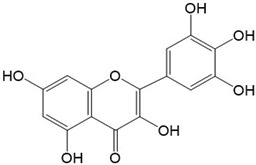
29. Epigallocatechin 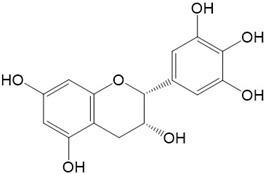	30. Tangeritin 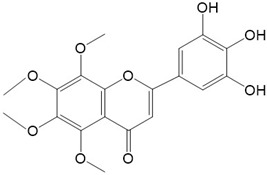
31. Rhamnetin 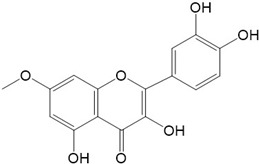	32. Pachypodol 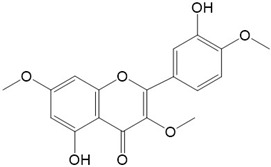
33. Homoeriodictyol 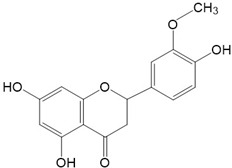	34. Aromadedrin 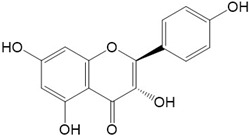
35. Glycitein 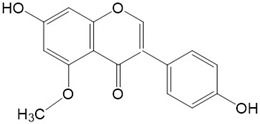	36. Pelargonidin 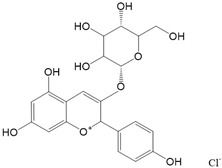

**Table 2 ijms-23-10020-t002:** Favorable effects of flavonoids against different AD targets in silico.

Class	Flavonoids	AD Targets	Ligand–Receptor Interactions	References
Flavanols	(−)-Epigallocatechin (EGC)	AChE	Hydrogen bonding	[82,83]
		BChE		
		GSK-3β		
		γ-secretase		[83]
		BACE-1		
	Epicatechin-3-O-gallate	AChE	Hydrogen bonding	[82]
		BChE		
	(−)-Epicatechin (EPC)	AChE	Hydrogen bonding	[83]
		BChE		
		GSK-3β		
		γ-secretase		
		BACE-1		
	(−)-Epigallocatechin gallate (EGCG)	AChE	Hydrogen bonding	[82,83]
Flavanols	(−)-Epigallocatechin gallate (EGCG)	BChE		
		GSK-3β		[83]
		γ-secretase		
		BACE-1		
	(+)-Catechin (CAT)	AChE	Hydrogen bonding	[82,83]
		BChE		
		GSK-3β		
		γ-secretase		[83]
		BACE-1		
	Artoflavanocoumarin	BACE-1	Hydrogen bonding; hydrophobic interactions	[84]
Flavanols	Epicatechin Gallate(ECG)	ApoE4	Hydrogen bonding; hydrophobic interactions	[85]
Flavanones	Cudraflavanone B	PTGS2	Hydrogen bonding	[86]
	Hesperidin	BACE-1	Hydrogen bonding	[87]
		AChE		
		BChE		
	Kolaviron	Aβ_42_ fibrils	Hydrogen bonding; hydrophobic interactions	[88]
	Macaflavanone C	GSAP	Hydrogen bonding; hydrophobic interactions	[89]
	Naringenin	AChE	Hydrogen bonding	[87]
	Pinocembrin	BACE-1	Hydrogen bonding	[90]
Flavanones	Pinostrobin	BACE-1	Hydrogen bonding	[90]
	Silibinin	AChE	Hydrogen bonding; π-π and π-H interaction	[91]
		Aβ42	Hydrophobic interactions	
	Taxifolin	α-amylase	Hydrogen bonding; π-π interaction	[92]
Flavones	Apigenin	BACE-1	Hydrogen bonding	[93]
		Aβ_42_ fibrils		[94]
	Baicalein	BACE-1	Hydrogen bonding; hydrophobic interactions	[95]
		AChE		
	Isovitexin	AChE	Hydrogen bonding	[96]
Flavones	Linarin	AChE	Hydrogen bonding; π-π interaction	[97]
	Vitexin	AChE	Hydrogen bonding	[96]
	Vitexin-4-O-glucoside	AChE	Hydrogen bonding	[96]
	Chrysin	AChE	Hydrogen bonding; π-π interaction; hydrophobic interactions	[98]
		BACE-1	Hydrogen bonding; hydrophobic interactions	[99]
		MAO-B		
Flavonols	2-(4′ Benzyloxyphenyl)-3-hydroxychromen-4-one	β-amyloid fibril	Hydrogen bonding	[100]
Flavonols	2-(4′ Benzyloxyphenyl)-3-hydroxychromen-4-one	β-amyloid	Hydrogen bonding; hydrophobic interactions	[100]
	8-Prenylkaempferol	PTGS2	Hydrogen bonding	[86]
	Icariin	AChE	-	[101]
		NMDAR	-	
		PDE5	-	
	Kaempherol	BACE-1	Hydrogen bonding	[93]
	Morin	Aβ_42_ protofibril	Hydrogen bonding; hydrophobic interactions; aromatic stacking interactions	[102]
Flavonols	Morin	BACE-1	Hydrogen bonding	[93]
Flavonols	Myricetin	BACE-1	Hydrogen bonding	[93]
	Quercetin	AChE	Hydrogen bonding	[103]
		BACE-1		[93]
		Aβ_42_ fibrils		[94]
	Fisetin	AChE	Hydrogen bonding	[104]
Isoflavones	Genistein	BACE-1	Hydrogen bonding	[105]
Isoflavones	Genistein	AChE	Hydrogen bonding; hydrophobic interactions	[106]
		BChE		
		NMDAR		
	Puerarin	AChE	Hydrogen bonding	[107]
		COX-2		
		C3		
		CaMK IIα		[108]

**Table 3 ijms-23-10020-t003:** Classification and promising preclinical studies of flavonoids and their neuroprotective role against Alzheimer’s disease.

Class	Flavonoids	Effects	Model	References
Flavanones	Naringin	Attenuates oxido-nitrosative stress and inflammation	ICV-STZ-induced rats	[109]
		Regulates multiple pathways, including amyloid β metabolism, tau protein hyperphosphorylation, acetylcholinergic system, glutamate receptor system, oxidative stress, and cell apoptosis	Hydrocortisone-induced mice	[110]
	Hesperetin	Multipotent effect, involving the inhibition of oxidative stress, and neuroinflammation	C57BL/6N mice treated with Aβ_1–42_	[111]
	Eriodictyol	Attenuates neuroinflammation and amyloidogenesis	LPS-induced C57BL/6J mice model and BV2 microglial cells	[112]
Flavanols	Epigallocatechin-3-gallate	Decreased the hyperphosphorylation of tau, suppressed BACE1 expression and activity as well as the expression of Aβ_1–42_, and promoted Ach content by diminishing the activity of AchE	AD rat models through an injection with Aβ _25–35_ solution	[113]
	Epigallocatechin	Alleviate Aβ_40_ aggregation and diminish ROS production, reduce the Aβ plaques in the brain	Neuroblastoma cells treated with Aβ_40_/APP/PS1 mouse	[114]
	(−)-Epicatechin	Reduces Aβ levels by inhibiting β, γ-secretase	TASTPM transgenic mouse model	[115]
		Inhibits tau phosphorylation	rTg4510 mouse model	[116]
Flavanols	Catechins	Decrease Aβ_42_ production, APP-C99/89 expression, γ-secretase component and Wnt protein levels, and γ-secretase activity, and increases the levels of APP-C83 protein and enzyme activities (α-secretase, neprilysin and Pin1)	NSE/hAPP-C105 Tg mice	[117]
Flavones	Luteolin	Decrease in the expression of Aβ_42_ aggregated, the oxidative stress, and apoptotic markers	Transgenic flies expressing human Aβ_42_ peptides	[118]
	Nobiletin	Improves cognitive impairment and reduces soluble Aβ levels	3xTg-AD mice model	[119]
		Reduces intracellular and extracellular β-Amyloid	iPS cells	[120]
Flavones	Diosmin	Reduces cerebral Aβ levels, tau hyperphosphorylation, and neuroinflammation	3xTg-AD mice model	[121]
	Apigenin	Preserves neuron and astrocyte morphology and reduces inflammation by regulating the expression of inflammatory mediators	LPS induced neuron/glial cells or neuron/glial cells treated with Aβ_1–42_	[122]
		Decreases the expression of GSK-3β with the consequence of lowering the hyperphosphorylation of tau protein and suppresses BACE1 expression	Wistar rats treated with Aβ _25–35_	[123]
Flavones	Wogonin	Attenuates amyloidogenic pathway by decreasing the levels of BACE1, APP β-C-terminal fragment, Aβ-aggregation, and phosphorylated tau	3xTg-AD mice model	[124]
	Chrysin	Attenuated Aβ-induced memory impairment through the reduction of lipid peroxidation levels and the elevation of antioxidant molecules	Sprague–Dawley rats treated with Aβ_25–35_	[125]
		Reverse learning impairment, reduced neuroinflammation induced by Aβ by lowering the expressions of IL-1, IL-10, and TNF-1 in the brain	Swiss mice treated with Aβ_1–42_	[126]
Flavonols	Kaempferol	Reduced the oxidative stress and acetylcholinesterase activity	Transgenic flies expressing human Aβ_42_ peptides	[127]
Flavonols	Quercetin	Reduces Aβ protein and tauopathy in hippocampus and amygdala	3xTg-AD mice model	[128]
	Morin	Ameliorates oxidative stress and neuroinflammation	Wistar rats treated with Aβ_1–42_	[129]
	Galangin	Decreases β-secretase, Aβ_42_, and p-tau levels; suppresses Beclin-1 and p-GSK3β expression; promotes p-Akt and p-mTOR expression	Okadaic-acid-induced PC12 cell	[130]
	Fisetin	Decreased the accumulation of Aβ, BACE-1 expression, and hyperphosphorylation of tau protein; increased the levels of both presynaptic and postsynaptic proteins	C57BL/6N mice treated with Aβ_1–42_	[131]
Anthocyanins	Cyanidin	Attenuates Aβ_25–35_-induced neuroinflammation	SK-N-SH cells (human neuroblastoma cell line) treated with Aβ_25–35_	[132]
	Pelargonidin	Inhibits of glial activation, cholinesterase, and oxidative stress	Wistar rats treated with Aβ_25–35_	[133]
		Decreases neuronal apoptosis	Wistar rats treated with Aβ_25–35_	[134]
Isoflavones	Genistein	Clears amyloid-β through PPARγ/ApoE activation	APPswe/PS1dE9 mice model	[135]
	Glycitein	Inhibits Abeta deposition and decreases oxidative stress	Caenorhabditis elegans (CL2006 and CL4176)	[136]
Isoflavones	Daidzein	Improves cognitive dysfunction and oxidative stress	ICV-STZ-induced rats	[137]
	Equol	Reduces Aβ-induced neurotoxicity via sustaining estrogen receptor alpha expression	SH-SY5Y cells treated with Aβ_25–35_	[138]
	7,3′,4′-Trihydroxyisoflavone	Suppresses the production of the proinflammatory mediators NO, iNOS, and COX-2 as well as of the proinflammatory cytokineIL-6 and inhibits reactive ROS generation	LPS-induced BV2 microglial cells	[139]

## Data Availability

Not applicable.

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
