# Peer review of "Protective Effects of Flavonoids against Alzheimer’s Disease: Pathological Hypothesis, Potential Targets, and Structure–Activity Relationship"

_ijms, 2022, doi:10.3390/ijms231710020_

Round 1
Reviewer 1 Report
The purpose of this manuscript is to describe the complex pathogenesis of Alzheimer’s disease, including all theories supported by the scientific community, and the possible role of flavonoids in counteracting this multifactorial pathology at several levels.
The scientific collect is very interesting, and each paragraph is described exhaustively. However, some problems, as indicated below, should be addressed before the document can be considered for publication in this journal. Here, I present all my objections in details.
Minor revision:
Line 12-87, unify the wording: Tau or tau
Line 179, the acronym TNF was already written in full in line 173
Line 259, “Tabel” should be modified in “Table”
Line 400, “euterpe oleracea martius” should be modified in “Euterpe oleracea martius”
It might be interesting to write a short paragraph that collects the pharmacological treatments used so far in the treatment of Alzheimer’s disease, specifying the mechanism of action, the side effects, and the reasons why their effectiveness is limited.
Could the authors combine Tables 2 and 3 into a single table? It might facilitate the reading of the manuscript.
Line 423, regarding the antioxidant effect of quercetin on the erythrocyte membrane, I would suggest adding these more recent references: (doi: 10.3390/ijms23147781 or PMID: 35887126; doi: 10.3390/molecules26164868 or PMID: 34443451).
Line 513-523, I would move this concept in the final conclusions.
Author Response
Point 1: Line 12-87, unify the wording: Tau or tau
Response 1: Thank you very much. We have unified the word as tau.
Point 2: Line 179, the acronym TNF was already written in full in line 173
Response 2: Thank you very much. We have corrected it.
Point 3: Line 259, “Tabel” should be modified in “Table”
Response 3: Thank you very much. We have modified it.
Point 4: Line 400, “euterpe oleracea martius” should be modified in “Euterpe oleracea martius”
Response 4: Thank you very much. We have corrected it.
Point 5: It might be interesting to write a short paragraph that collects the pharmacological treatments used so far in the treatment of Alzheimer’s disease, specifying the mechanism of action, the side effects, and the reasons why their effectiveness is limited.
Response 5: Thanks very much for the reviewer’s good advice. We have added a short paragraph that describe the pharmacological treatments used so far in the treatment of Alzheimer’s disease in the text. And please see line 244-255 in text for details.
Point 6: Could the authors combine Tables 2 and 3 into a single table? It might facilitate the reading of the manuscript.
Response 6: Table 2 and Table 3 illustrate the neuroprotective role of flavonoids against Alzheimer’s disease from different perspectives. Table 2 is the data in silico and Table 3 is the date in AD model. We think that putting them all together will mess up these data.
Point 7: Line 423, regarding the antioxidant effect of quercetin on the erythrocyte membrane, I would suggest adding these more recent references: (doi: 10.3390/ijms23147781 or PMID: 35887126; doi: 10.3390/molecules26164868 or PMID: 34443451).
Response 7: Thank you very much. We have added these recent references in the corresponding position in the text.
Point 8: Line 513-523, I would move this concept in the final conclusions.
Response 8: Thank you very much. This paragraph is the summary of the part “Flavonoids”. We think if it is put into the final conclusions of whole article, it will be somewhat repetitive and uncoordinated with the content in the final conclusions. And the part of “Flavonoids” will be incomplete.
Reviewer 2 Report
The presented manuscript presents a review on the flavonoids, which exhibit multiple biological activities and anti-Alzheimer disease effects and could be a part of multitarget prevention and treatment strategies for AD. Flavonoids are a broad and prominent group of secondary metabolites, so far the most widely analyzed. Thus, this group's choice, also in the context of their spread in herbal medicines and bioavailability for humans, seems the most interesting. As the Authors stressed in the conclusion, the described studies are mainly in vitro analyses, but they could serve as the basis for future clinical studies. I read the manuscript thoroughly and cannot address any serious drawbacks of the work. The manuscript seems suitable for publication in the International Journal of Molecular Sciences.
- The description of the disease pathogenesis origin is extensive.. The proportion of this part and the characterization of flavonoids anti-AD actions seems inappropriate. I would suggest condensing the part of the manuscript described on pages 3-5.
- The structures of flavonoids in table 2 should be smaller, proportional – now the differences in the size of flavonoids ring are too significant- and placed in order.
Author Response
Point 1:The description of the disease pathogenesis origin is extensive. The proportion of this part and the characterization of flavonoids anti-AD actions seems inappropriate. I would suggest condensing the part of the manuscript described on pages 3-5.
Response 1: Thank you very much. We have modified the part of the manuscript described on pages 3-5.
Point 2: The structures of flavonoids in table 2 should be smaller, proportional – now the differences in the size of flavonoids ring are too significant- and placed in order.
Response 2: Thank you very much. We have modified the structures of flavonoids and placed them in order.